# Attitudes towards Communication in Nursing Students and Nurses: Are Social Skills and Emotional Intelligence Important?

**DOI:** 10.3390/healthcare11081119

**Published:** 2023-04-13

**Authors:** Lucía Sanchis-Giménez, Laura Lacomba-Trejo, Vicente Prado-Gascó, María del Carmen Giménez-Espert

**Affiliations:** 1Psychology Undergraduate Student, Faculty of Psychology and Speech Therapy, Universitat de València, 46010 Valencia, Spain; sangilu@alumni.uv.es; 2Department of Social Psychology, Faculty of Psychology and Speech Therapy, Universitat de València, 46010 Valencia, Spain; laura.lacomba@uv.es (L.L.-T.); vicente.prado@uv.es (V.P.-G.); 3Nursing Department, Faculty of Nursing and Chiropody, Universitat de València, 46010 Valencia, Spain

**Keywords:** attitudes towards communication, emotional intelligence, social skills, nursing students, nurses

## Abstract

The communication attitude (ACO) of nurses can significantly influence patient health outcomes. This work aims to evaluate predictor variables of communication attitude (emotional intelligence and social skills) in nurses and nursing students separately by comparing linear and non-linear methodologies. Two samples participated in this study: 312 nursing professionals and 1369 nursing students. In total, 75.60% of the professionals and 83.80% of the students were women. After signing the informed consent form, their emotional intelligence (TMMS-24), social skills (IHS) and ACO (ACO) were assessed. It was found that ACO through linear regression models in professionals was predicted by emotional repair and, in students, by attention and emotional repair, as well as by low exposure to new situations, low social skills in the academic or work area and high empathy. Overall, the comparative qualitative models show how the combination of different skills related to emotional intelligence and social skills lead to high levels of ACO. Conversely, their low levels result in an absence of ACO. Our results highlight the importance of emotional intelligence, especially emotional repair and empathy, as well as the need to consider ways to encourage the learning of these skills in a formal way.

## 1. Introduction

Communication between healthcare professionals and the patient is influenced by the healthcare professional’s model of care and is central to the health and disease process [1]. The bio-psychosocial model [2,3] maintains that patient-centred communication is essential to achieve better health outcomes [4], including greater involvement in treatment, improved quality of life and reduced use of health services [5,6].

In this sense, nurses are one of the agents who interact most frequently with patients; their role is key to establishing effective communication between the patient, family and other team members, with communication being one of the most important indicators for assessing the quality of care [7]. Through effective communication, informed decision-making, care, autonomy, adherence, survival and involvement in treatment are enhanced and, therefore, patient health outcomes are improved [5,6,8]. In line with the above, poor patient–health professional communication is associated with poorer treatment management, more avoidable adverse events and higher mortality rates [9,10,11].

The literature suggests that nurse–patient communication is influenced by situational and dispositional factors [12]. The former factor is related to the conditions of the healthcare system, which are generally not controllable by nurses [13]; the latter factor is related to personal variables, including social skills, empathy and emotional intelligence [14,15,16].

Thus, emotional intelligence is understood as a person’s ability to perceive, attend to, regulate and clarify their emotions and those of other people [17]. In turn, nurses with a greater capacity for emotional management or emotional intelligence show more effective, open, empathic and genuine communication [18]. In relation to the above, this ability is associated with a more effective communicative attitude, where clearer information is given about treatments and their consequences and there is active and empathetic listening [19,20,21,22].

On the other hand, social skills are learned behaviours that facilitate interaction with others in the social and work environment in an effective manner [23]. The social skills of nursing professionals include their ability to communicate effectively and provide high-quality care [24]. This factor is associated with better physical and emotional health outcomes for patients, such as increased adherence to treatment and improved survival; it is also associated with more social support, improved quality of life and fewer mental health problems in nurses [24,25,26,27]. Therefore, they are essential for the personal and professional development of nurses. In this context, empathy, as part of social skills, is especially important [28]. Empathy is the ability to understand other people’s emotions, thoughts and feelings and to put oneself in their place. Empathetic nurses are aware of patients’ needs and reasons, communicate openly and sympathetically with them and foster their autonomy and build positive self-concepts related to their ability to manage treatment [29]. Consequently, they have a positive attitude towards communication, leading to a higher-quality communication process, which generates more effective support relationships [14,15] and improved health outcomes.

Furthermore, research suggests that nursing care and patient outcomes and safety are directly related to nurses’ personal competencies [27]. As can be seen, all these skills are of great importance in nursing, which is why they are fundamental in the preparation of nursing students as future professionals. Training to enhance these skills is essential for effective communication, as well as to increasing well-being, self-care and positive stress management among nurses [7,30,31]. In this sense, it is necessary to know the mechanisms that explain communication attitudes in nurses and nursing students.

Despite the above issues, few studies have been conducted that assess whether emotional intelligence and social skills predict attitudes towards communication in nurses and nursing students. Nurses’ skills have been shown to be fundamental in the care of patients and family members and even influence the survival chances of patients [5,6,8]. There is a gap in the literature in this regard; these variables have not been studied jointly, while the profiles of students and professionals have been analyzed separately. Even fewer studies have used linear and non-linear methods of prediction. For these reasons, this paper aims to complete the following tasks: evaluate the predictor variables of communication attitude in nurses and nursing students separately; and find out whether the variables predicting communicative attitude are different in practicing nurses and nursing students. Based on previous literature, (H1) predicts that emotional intelligence and social skills, especially empathy, predict attitude towards communication in nurses. Similarly, (H2) predicts that lack of emotional intelligence and social skills, especially empathy, predict the absence of attitude towards communication in nurses.

## 2. Materials and Methods

### 2.1. Participants

Two types of sample participated in this study: 312 nursing professionals and 1369 nursing students. In total, 75.60% (*n* = 236) of the professionals were women and 24.40% (*n* = 76) were men. The mean age of the professionals was 44.41 years (SD = 11.09; rank = 22 to 64 years). In reference to educational level of the professionals, the majority had a bachelor’s degree (79.80%), 17.70% had a master’s degree and 2.40% had a doctorate degree. Regarding employment status of the professionals, 53.80% had permanent positions, 28.40% had a substitute position and 17.80% had a temporary contract. On the other hand, 83.80% (*n* = 1147) of the students were women and 16.20% (*n* = 222) were men. In this sample, the mean age was 21.79 years (SD = 5.34; rank = 17 to 55 years).

### 2.2. Measures

The study variables are divided in three different groups: emotional intelligence, social skills and attitudes towards communication.

#### 2.2.1. Emotional Intelligence

These variables were evaluated with the trait emotional meta-mood scale (TMMS-24), which was adapted by Fernández-Berrocal et al. [32] based on the original version trait meta-mood scale [33]. It has 24 items divided in three dimensions: emotional attention, emotional clarity and emotional repair. A five-point Likert scale was used ranged from 1 = strongly disagree to 5 = strongly agree. Thus, higher scores in the three dimensions equated to high emotional intelligence. This instrument has showed adequate psychometric properties in previous studies [34] and did so in this study.

#### 2.2.2. Social Skills

These variables were assessed with the social skills inventory (IHS), an adaptation by Olaz et al. [35] based on the social skills inventory [36]. It has 26 items divided in five dimensions: conversation and social performance, self-exposure to strangers and new situations, empathic skills and expression of positive feelings, coping with risk, social, academic and work. It includes a five-point Likert scale ranged from 0 = never or rarely to 4 = always or most always. The conversation and social performance scale were inverted. Thus, higher scores in the rest of the dimensions equated to high social skills. This instrument has shown adequate psychometric properties in previous studies [35] and did so in this study.

#### 2.2.3. Attitudes towards Communication

This variable was evaluated with the instrument attitudes towards communication of nurses (ACO) [37] (intellectual property registered at the University of Valencia on 8 April 2019, registration number: UV-MET-201917R). An adaptation for nursing students was used for the student’s sample [38]. It includes 25 items divided in three dimensions: affective, cognitive and behavioural. A five-point Likert was used ranged from 1 = strongly disagree to 5 = strongly agree. Both instruments show adequate psychometric properties [37] and did so in this study.

### 2.3. Procedure

Both professionals and students were selected through convenience sampling.

#### 2.3.1. Professionals (Sample 1)

Data in the first sample was collected for the period September 2015–February 2016. The nursing professionals were recruited from six Spanish hospitals in the Valencian Community. They had to answer the instruments on site, spending approximately 35 min doing so.

#### 2.3.2. Students (Sample 2)

Data for the second sample were collected during the 2018/2019 academic year. The sample of nursing students was collected from five universities in the Valencian Community (Spain), of which two were public entities and three were private entities. The students were enrolled on the universities’ nursing degree programs, which gave their consent for participation. They completed the instruments in the classroom; this stage lasted for around 35 min. It is relevant to mention that this study was authorized by the Research Ethics Committee of the university where the study was performed (H1529396558647).

### 2.4. Data Analysis

Firstly, descriptive statistics were calculated. Secondly, a hierarchical regression analysis in two steps: (1) we added the emotional intelligence variables and the social skills variables; and (2) the fuzzy-set Qualitative Comparative Analisys (fsQCA) statistical analysis was performed. The descriptive analysis and the hierarchical regression model were calculated with Statistical Package for the IBM SPSS Statistics 24 (IBM Corporation, Armonk, NY, USA) and QCA models with fsQCA 2.5 [39,40].

### 2.5. Study Design

The study design was cross-sectional.

## 3. Results

### 3.1. Descriptive Analysis

Descripted analysis and calibration values for the variables used on fsQCA in Sample 1 are presented in Table 1. The same statistics for Sample 2 are shown in Table 2. Both show the mean, standard deviation, range and percentiles.

### 3.2. Hierarchical Regression Model

We conducted a linear regression to predict communication attitudes in each of the samples (nursing students and professionals) (Table 3).

#### 3.2.1. Professionals (Sample 1)

In the prediction of ACO, two steps were established (R^2^_adjusted_ = 0.03; *p* = 0.43). Firstly, all dimensions of TMMS-24 were entered and the results showed that emotional intelligence variables explained 4% of variance (∆R^2^_adjusted_ = 0.04; *p* ≤ 0.01); emotional repair was the only variable that had a significant positive effect on the ACO (*β* = 0.19; *p* ≤ 0.01). Secondly, social skills variables were included (∆R^2^_adjusted_ = 0.011; *p* = 0.43). Considering each of the dimensions, emotional repair continued to be the only significant predictor of ACO (*β* = 0.20; *p* ≤ 0.01).

#### 3.2.2. Students (Sample 2)

In the prediction of ACO, the same steps were established for the students sample (R^2^_adjusted_ = 0.03; *p* ≤ 0.001). Emotion intelligence variables explained 1% of variance (∆R^2^_adjusted_ = 0.01; *p* ≤ 0.001). In this model, results showed that emotional attention (*β* = 0.08; *p* ≤ 0.01) and emotional repair (*β* = 0.06; *p* ≤ 0.05) were significant positive predictors. Social skills dimensions were then added to the model (∆R^2^_adjusted_ = 0.03; *p* ≤ 0.001). Considering each of the variables, emotional attention (*β* = 0.06; *p* ≤ 0.05) and emotional repair (*β* = 0.08; *p* ≤ 0.05) continued to be significant positive predictors. In this second step, empathic skills and expression of positive feelings (*β* = 0.13; *p* ≤ 0.001) was shown to be a positive predictor. On the other hand, self-exposure to strangers and new situations (*β* = −0.11; *p* ≤ 0.01) and academic social and work (*β* = −0.06; *p* ≤ 0.05) were significant negative predictors of ACO.

### 3.3. Fuzzy-Set Qualitative Comparative Analysis (fsQCA)

After calculating the hierarchical regression model, fsQCA statistics were carried out. The raw data responses were transformed into fuzzy-set responses. Firstly, all the cases with missing data were deleted. Secondly, all constructs were obtained by calculating the average obtained in the items that make up each construct. Thirdly, values of the conditions were calibrated using three thresholds [41]: percentile 10% (low levels; fully outside the set), 50% (intermediate levels; neither inside nor outside the set), and 90% (high levels; fully inside the set) [42]. Finally, necessary analyses (Table 4) and sufficiency analyses (Table 5) were calculated for both samples.

#### 3.3.1. Necessary Conditions

According to the necessary conditions assessment (Table 4), there were no necessary conditions for high or low values of ACO as there was no consistency higher than 0.90 in Sample 1. The same results were shown in sample 2.

#### 3.3.2. Sufficient Conditions

Sufficiency analyses were conducted in both samples (Table 5). As for the professionals sample, the intermediate solution indicated 19 combinations of causal conditions that explained 54% (overall coverage = 0.54; overall consistency = 0.80) of the high levels of ACO. The most important sufficient conditions were the interaction of high levels of emotional attention, empathic skills and expression of positive feelings and academic social and work, and low levels of emotional clarity (raw coverage = 0.25; consistency = 0.87), which explained 25% of the cases with a high score in ACO. However, nine combinations were obtained that explained 38% of the low levels of ACO (overall coverage = 0.38; overall consistency = 0.82). Considering all the different combinations, the most important ones were the interaction of high levels of emotional attention, high levels of empathic skills and expression of positive feelings and academic social and work, and low levels of emotional repair and conversation and social performance (raw coverage = 0.22; consistency = 0.86), which explained 22% of the cases with a low score in ACO.

As for the students sample, 23 combinations were indicated and explained 47% (overall coverage = 0.47; overall consistency = 0.80) of the high levels of ACO. The most important sufficient conditions obtained were the interaction of high levels of emotional clarity, empathic skills and expression of positive feelings and coping with risk (raw coverage = 0.21; consistency = 0.86) that explained 21% of the cases with high scores of ACO. Meanwhile, for the prediction of low levels of ACO, nine combinations were shown that explained 45% of the low scores (overall coverage = 0.45; overall consistency = 0.80). The most important sufficient conditions were the interaction of high levels of emotional repair, self-exposure to strangers and new situations and academic social and work (raw coverage = 0.22; consistency = 0.85), which explained 22% of the cases with low levels of ACO.

## 4. Discussion

This study aimed to find out which skills influence ACO in nurses and nursing students through the comparison of linear and non-linear models. It also aimed to find out whether the variables that explain ACO are different for practicing nurses and student nurses. Overall, it is observed that different emotional intelligence skills contribute differently to the presence of communicative attitudes in nurses. Similarly, the role of social skills seems to be very relevant, especially empathy.

We sought to answer (H1), which stipulated that emotional intelligence and social skills, especially empathy, could predict the attitude towards communication of nurses, and (H2), which stated that the absence of these skills would predict the absence of attitude towards communication of nurses. Through the linear models it is noted that in the case of professional women, only emotional repair predicted attitude towards communication. In student nurses emotional care and repair and empathy, together with low exposure to new situations and low social skills in the academic environment, predicted attitude towards communication. It is striking that emotional attention is a predictor of emotional skills since generally extreme levels (high or low) of this variable have been associated with the presence of mental health problems [44]. However, other studies have shown how emotional attention in interaction with empathy leads to high levels of self-esteem and emotional well-being [45].

Nevertheless, it should be noted that the linear models were not practically explanatory. Despite showing statistically significant predictive models, the percentage of variance explained in both professional and student nurses were very low (less than 10%). These results encourage us to consider that perhaps the explanation of such a complex phenomenon as attitude towards communication needs further analysis. In particular, it would be interesting to explore the influence of other organizational variables, such as knowledge self-efficacy, specific training of the participants, type of resource (public, subsidized, private), availability of treatment technology, type of contract, job satisfaction or school and work environment, on the attitude towards communication in nursing [27,46,47].

Thus, it may be interesting to consider the results of the comparative qualitative analyses in a complementary way. Through these models, the different combinations of variables predict between 47 and 54% of the cases of high attitude towards communication and between 38 and 45% of the cases of no attitude towards communication. Specifically, in the case of nursing professionals, 19 possible combinations were obtained, predicting 54% of the variance. The most relevant sufficient combination was the interaction between high levels of emotional attention, empathy and expression of positive feelings and social skills in work environments, and low emotional clarity.

The second and third most relevant pathways take into account the combination of the same variables with emotional repair and coping with risk. The above seems to indicate that, for nursing professionals, emotional repair and empathy are key variables in predicting attitude towards communication. These variables operate in combination with the presence or absence of social skills at work, coping hit risk and emotional clarity. Previous work in this profession has pointed out the importance of emotional regulation and empathy for care giving, as well as for ACO [15,22,48]. These results are especially encouraging because they indicate that modifiable variables, such as social skills or emotional intelligence, can influence attitudes towards communication.

In the case of low levels of ACO in nurses, it can be observed that nine combinations predicted 38% of the cases. Therefore, the most important pathway resulted from communication between high levels of emotional attention, empathic skills and expression of positive feelings and academic social and work, and low levels of emotional repair and conversation and social performance. In all three main pathways, the absence of conversation and social performance and the presence of social skills at work were relevant variables. Their interaction with the absence of emotional repair or clarity was crucial. Thus, despite showing social skills in the work environment, the absence of conversation and social performance and emotional repair lead to low levels of ACO. In this sense, the absence of conversation and social performance or social skills, and emotional repair or emotional management and regulation, generates low levels of ACO, as previous work has pointed out [16,18,19]. In view of the above, it could be inferred that, in the case of practising nurses, programs that enhance emotional regulation and empathy could be useful to enhance their ACO. Therefore, it would be interesting to establish intervention programs for nurses in hospital centres.

In the case of nursing students, 23 possible combinations of explanations for high levels of ACO were obtained. These predicted 47% of the cases. The most important sufficient conditions obtained were the interaction of high levels of emotional clarity, empathic skills and expression of positive feelings and coping with risk. In the three main combinations, the presence of empathy and expression of positive feelings, coping with risk and emotional clarity, in interaction with the absence of conversation and social performance and self-exposure to strangers and new situations, resulted in high levels of ACO. Thus, empathy and the expression of positive feelings is again central; however, in the case of students, of equal importance is their ability to cope with risk situations.

In reference to the absence of attitudes towards communication in nursing students, nine pathways were found that predicted 45% of the cases. The most relevant were the interaction of high levels of emotional repair, self-exposure to strangers and new situations and academic social and work. In the second and third pathways, it is observed that despite presenting high levels of emotional competencies (attention, clarity or emotional repair) and empathy or social skills in the work environment, their interaction with the absence of conversation and social performance or coping with risk results in low ACO.

These data seem surprising because they point to the importance of social skills over emotional intelligence in nursing students. For this reason, for nursing students it is necessary to establish an intervention that enhances their confidence and ability to cope with risk situations, as well as their social skills. Empowering emotional intelligence in nurses can reduce burnout and psychosocial risks and increase job satisfaction. This will lead to better personal and health outcomes [49]. It is considered that at least one subject should be clearly established within the nursing degree where the empowerment of the personal skills mentioned above should be worked on. Activities such as video-stimulated recall, role playing or working through case studies can be helpful [30,50,51,52]. These skills are worked on in a transversal manner throughout the nurses’ training. However, they should be formally and explicitly included in the nursing degree. These subjects should also be evaluated in a practical way and could be taught by psychologists and nurses with specific training in this regard.

Lack of ACO from nurses to patients can lead to poorer health outcomes. Patients may feel uninformed and disempowered by lack of understanding; their adherence to treatment may also be compromised. In fact, ACO are one of the most valued indicators of quality of care and even have repercussions on patient morbidity and mortality [21,30,31]. Training in emotion management, social skills and empathy in nurses and students can be a simple and inexpensive tool that can have great economic, social, occupational, personal and health repercussions [30,53].

Despite the great contributions, our work has some limitations. On one hand, the research design and the size of the sample complicate the generalization of the data to the Spanish population. Our data may only be representative of the situation in the Valencian Community. In addition, the different sample collection period between professionals and students could be a limitation. However, since the variables under study refer to personal interactions in the work system, they may not be badly affected by the passage of time. Furthermore, future studies should carry out longitudinal designs to evaluate whether the COVID-19 pandemic has modified the factors that are relevant to ACO in these groups. On the other hand, other studies could evaluate variables, such as resilience or coping skills, which are also relevant in these contexts. However, the results of the study can be considered a first approximation to improve the training of nurses.

## 5. Conclusions

In conclusion, emotional intelligence, social skills, particularly empathy and the ability to cope with risk situations, are fundamental skills in the care process in a healthcare setting. Attitudes towards nurse–patient communication may be influenced by them. Formal training should take this information into account to promote these skills in the educational system and, thus, generate future professionals capable of providing person-centred care. Likewise, in the case of practicing professionals, governments should consider including training that enhances these skills to improve care and quality of care. Thus, a proposal based on the results obtained would be to include the enhancement of emotional, social and communication skills as part of the formal education curriculum. As for active nurses, an online or face-to-face intervention program could be carried out to enhance these skills. There are some interesting proposals in this regard that are based on improving their mental health and communication skills and reducing burnout [53,54,55].

## 6. Patents

The instrument attitudes towards communication of nurses (ACO) [37] was registered as intellectual property belonging to the Universitat de València on 8 April 2019 (registration number: UV-MET-201917R).

## Figures and Tables

**Table 1 healthcare-11-01119-t001:** Descriptive statistics for professionals.

	Variables	Mean	SD	Range	Percentile		
					10	50	90
EI	Emotional attention	3.45	0.68	1–5	2.62	3.42	4.37
	Emotional clarity	3.83	0.69	1–5	3.00	3.87	4.87
	Emotional repair	3.93	0.68	1–5	3.00	4.00	4.87
SS	Conversation and social performance	2.17	0.74	1–5	1.28	2.00	3.24
	Self-exposure to strangers and new situations	3.32	0.74	1–5	2.50	3.33	4.33
	Empathic skills and expression of positive feelings	4.37	0.71	1–5	3.33	4.66	5.00
	Coping with risk	3.55	0.77	1–5	2.50	3.50	4.50
	Social academic and work	3.52	0.84	1–5	2.33	3.66	4.66
ACO	Attitudes towards communication	3.20	0.38	1–5	2.84	3.20	3.56

Note: EI: emotional intelligence dimensions; SS: social skills dimensions; ACO: attitudes towards communication; SD: standard deviation.

**Table 2 healthcare-11-01119-t002:** Descriptive statistics for students.

	Variables	Mean	SD	Range	Percentile		
					10	50	90
EI	Emotional attention	3.62	0.71	1–5	2.73	3.62	4.50
	Emotional clarity	3.54	0.74	1–5	2.50	3.62	4.50
	Emotional repair	3.69	0.72	1–5	2.75	3.75	4.62
SS	Conversation and social performance	2.42	0.76	1–5	1.42	2.28	3.42
	Self-exposure to strangers and new situations	3.24	0.73	1–5	2.33	3.16	4.20
	Empathic skills and expression of positive feelings	4.33	0.67	1–5	3.33	4.33	5.00
	Coping with risk	3.54	0.79	1–5	2.50	3.50	4.50
	Social academic and work	3.24	0.87	1–5	2.00	3.33	4.33
ACO	Attitudes towards communication	3.19	0.38	1–5	2.88	3.16	3.60

Note: EI: emotional intelligence dimensions; SS: social skills dimensions; ACO: attitudes towards communication; SD: standard deviation.

**Table 3 healthcare-11-01119-t003:** Hierarchical regression model.

		Criterion Variables			
		Professionals		Students	
		ACO		ACO	
Predictors		∆R^2^	*β*	∆R^2^	*β*
Step 1		0.04 **		0.01 **	
	Emotional attention		0.07		0.08 **
	Emotional clarity		−0.06		−0.05
	Emotional repair		0.19 **		0.06 *
Step 2		0.01			
	Emotional attention		0.05		0.06 *
	Emotional clarity		−0.05		−0.01
	Emotional repair		0.20 **		0.08 *
	Conversations and social performance		0.09		0.04
	Self-exposure to strangers and new situations		−0.01		−0.11 **
	Empathic skills and expression of positive feelings		0.05		0.13 ***
	Coping with risk		−0.04		0.01
	Academic social and work		0.05		−0.06 *
Total					
R^2^_adj_		0.03		0.03 **	

Note: ∆R^2^ = change on R^2^; R^2^_adj_ = R^2^_adjusted_; *β* = regression coefficient; * *p* < 0.05; ** *p* ≤ 0.01; *** *p* ≤ 0.001.

**Table 4 healthcare-11-01119-t004:** Necessary conditions.

	Professionals				Students			
	ACO		~ACO		ACO		~ACO	
	Cons	Cover	Cons	Cover	Cons	Cover	Cons	Cover
Emotional attention	0.63	0.67	0.58	0.56	0.64	0.63	0.62	0.59
~Emotional attention	0.58	0.61	0.65	0.61	0.59	0.62	0.62	0.62
Emotional clarity	0.61	0.67	0.60	0.59	0.59	0.62	0.63	0.63
~Emotional clarity	0.63	0.64	0.67	0.60	0.65	0.65	0.62	0.60
Emotional repair	0.65	0.69	0.58	0.56	0.61	0.62	0.63	0.62
~Emotional repair	0.59	0.61	0.68	0.63	0.63	0.64	0.61	0.60
CO	0.62	0.65	0.63	0.59	0.67	0.65	0.62	0.58
~CO	0.61	0.65	0.63	0.60	0.57	0.61	0.63	0.65
SE	0.60	0.65	0.60	0.59	0.59	0.58	0.67	0.64
~SE	0.62	0.63	0.65	0.59	0.64	0.67	0.57	0.57
EM	0.63	0.66	0.60	0.57	0.70	0.62	0.68	0.57
~EM	0.59	0.62	0.64	0.61	0.51	0.62	0.55	0.64
Coping with risk	0.66	0.65	0.65	0.58	0.63	0.61	0.66	0.61
~Coping with risk	0.57	0.64	0.60	0.61	0.60	0.65	0.58	0.60
SO	0.62	0.68	0.58	0.58	0.59	0.61	0.64	0.63
~SO	0.61	0.62	0.68	0.62	0.64	0.65	0.61	0.59

Note: ACO: attitudes towards communication; ~: absence of condition; Cons: consistency; cover: coverage; CO: conversation and social performance; SE: self-exposure to strangers and new situations; EM: empathic skills and expression of positive feelings; So: social academic and work; condition needed: consistency ≥ 0.90.

**Table 5 healthcare-11-01119-t005:** Three main sufficient conditions for ACO and for absence of ACO.

Frecuency Cutoff: 1	Professionals						Students					
	ACO			~ACO			ACO			~ACO		
	Consistency Cutoff: 0.89			Consistency Cutoff: 0.89			Consistency Cutoff: 0.89			Consistency Cutoff: 0.88		
	1	2	3	1	2	3	1	2	3	1	2	3
Emotional attention	●			●		●		●			●	
Emotional clarity	⚬	●	⚬		⚬		●		●		●	
Emotional repair		●	●	⚬	⚬				⚬	●		●
CO				⚬	⚬	⚬	⚬	⚬			⚬	⚬
SE						⚬	⚬	⚬		●	●	
EM	●	⚬	●	●	●	⚬	●	●	●			●
Coping with risk		⚬	●		⚬		●	●	●	⚬		⚬
SO	●		⚬	●	●	●			●	●	⚬	●
Raw coverage	0.25	0.21	0.21	0.22	0.19	0.18	0.21	0.20	0.20	0.22	0.21	0.21
Unique coverage	0.01	0.02	0.01	0.03	0.01	0.02	0.00	0.01	0.01	0.00	0.02	0.01
Consistency	0.87	0.85	0.87	0.86	0.90	0.84	0.86	0.86	0.86	0.85	0.85	0.84
Overall solution consistency			0.80			0.82			0.80			0.80
Overall solution coverage			0.54			0.38			0.47			0.45

Note: ACO: attitudes towards communication; ~: absence of condition; CO: conversation and social performance; SE: self-exposure to strangers and new situations; EM: empathic skills and expression of positive feelings; SO: social academic and work; ● = presence of condition, ⚬ = absence of condition; expected vector for ACO (1, 1, 1, 1, 1, 1, 1, 1), for ~ACO (0, 0, 0, 0, 0, 0, 0, 0) using the format of Fiss [43].

## Data Availability

The database used in this work is available on request to the corresponding author.

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
