# Peer review of "Attitudes towards Communication in Nursing Students and Nurses: Are Social Skills and Emotional Intelligence Important?"

_healthcare, 2023, doi:10.3390/healthcare11081119_

Round 1
Reviewer 1 Report
This works aims to evaluate predictor variables of communication attitude (emotional intelligence and social skills) among nursing students and nursing professionals, separately. This study going to high level, however editing needed. Authors need to build a stronger case of why need to do this study.
Introduction:
Authors need to build a stronger case the motivation of doing this study (Clear problem statement needed). See this reference might help you.
· Lake, E.T., Riman, K.A. and Lee, C.S., 2022. The association between hospital nursing resource profiles and nurse and patient outcomes. Journal of Nursing Management, 30(3), pp.836-845.
Line 34: "maintains that patient-centred communication is essential to achieve better health outcomes". This statement needs to be supported by previous study. See this might help.
· Jarrar MT, Al-Bsheish M, Aldhmadi BK, Albaker W, Meri A, Dauwed M, Minai MS. Effect of practice environment on nurse reported quality and patient safety: the mediation role of person-centeredness. InHealthcare 2021 Nov (Vol. 9, No. 11, p. 1578). Multidisciplinary Digital Publishing Institute.
Line 43: See this One more study investigate the relationship between emotional intelligence and nurses’ awareness towards pain management, the results revealed a significant positive relationship between nurses’ emotional intelligence and their pain management awareness. This to be used to strengthen your study case.
· Issa MR, Muslim NA, Alzoubi RH, Jarrar MT, Alkahtani MA, Al-Bsheish M, Alumran A, Alomran AK. The Relationship between Emotional Intelligence and Pain Management Awareness among Nurses. InHealthcare 2022 Jun 4 (Vol. 10, No. 6, p. 1047). MDPI.
Methods:
Provide a heading of study design.
Sampling procedure in Line 122, is not clear how six Spanish hospitals were selected. Furtermore, the data collected from professionals were outdate. Need to provide an approriate justification. In the first suggested reference of this review can find in page 13 the study limitations provided an justification that “Because the study variables are interpersonal interactions in a work system, they are less likely to be affected by time.” See the limitations part of “Effect of practice environment on nurse reported quality and patient safety: the mediation role of person-centeredness”.
In Data Analysis need to write the Abbreviations in full first time presented such as “QCA” or “fsQCA”.
Results:
Remove the explanations of Methods from results section. Move these to methodology.
Discussions:
Line 253: "it would be 251 interesting to explore the influence of other organizational variables such as type of con-252 tract, job satisfaction or school and work environment on the attitude towards communi-253 cation in nursing [42]"
Other factors also could knowledge self-efficacy, see this article :
· Shehab S, Al-Bsheish M, Meri A, Dauwed M, Aldhmadi BK, Kareem HM, Alsyouf A, Al-Mugheed K, Jarrar MT. Knowledge sharing behaviour among head nurses in online health communities: The moderating role of knowledge self-efficacy. PloS one. 2023 Jan 19;18(1):e0278721.
Author Response
Dear editor and reviewers,
Thanking you in advance for your work, time and effort, and responding to your request and that of the reviewers, we are sending you the manuscript again. In it you can see in blue everything we have modified.
In summary, we have included more theoretical information in the introduction section and modified general aspects of the language. We have also provided more information in the method and discussion section and future lines of research.
Reviewer 1.
This works aims to evaluate predictor variables of communication attitude (emotional intelligence and social skills) among nursing students and nursing professionals, separately. This study going to high level, however editing needed. Authors need to build a stronger case of why need to do this study.
Introduction:
Authors need to build a stronger case the motivation of doing this study (Clear problem statement needed). See this reference might help you.
Following the reviewers' recommendations, the information has been expanded and all the studies suggested by the reviewer have been added in the section on future lines of research (Page 2, lines 61, 72-73, 81 – 85).
Lake, E.T., Riman, K.A. and Lee, C.S., 2022. The association between hospital nursing resource profiles and nurse and patient outcomes. Journal of Nursing Management, 30(3), pp.836-845.
Line 34: "maintains that patient-centred communication is essential to achieve better health outcomes". This statement needs to be supported by previous study. See this might help.
- Jarrar MT, Al-Bsheish M, Aldhmadi BK, Albaker W, Meri A, Dauwed M, Minai MS. Effect of practice environment on nurse reported quality and patient safety: the mediation role of person-centeredness. InHealthcare 2021 Nov (Vol. 9, No. 11, p. 1578). Multidisciplinary Digital Publishing Institute.
Line 43: See this One more study investigate the relationship between emotional intelligence and nurses’ awareness towards pain management, the results revealed a significant positive relationship between nurses’ emotional intelligence and their pain management awareness. This to be used to strengthen your study case.
- Issa MR, Muslim NA, Alzoubi RH, Jarrar MT, Alkahtani MA, Al-Bsheish M, Alumran A, Alomran AK. The Relationship between Emotional Intelligence and Pain Management Awareness among Nurses. InHealthcare 2022 Jun 4 (Vol. 10, No. 6, p. 1047). MDPI.
Following the reviewers' recommendations, the information has been expanded and all the studies suggested by the reviewer have been added in the section on future lines of research (Page 2, line 45).
Methods:
Provide a heading of study design.
Following the reviewers' recommendations, this section has been included (Page 4, line 155).
Sampling procedure in Line 122, is not clear how six Spanish hospitals were selected. Furtermore, the data collected from professionals were outdate. Need to provide an approriate justification. In the first suggested reference of this review can find in page 13 the study limitations provided an justification that “Because the study variables are interpersonal interactions in a work system, they are less likely to be affected by time.” See the limitations part of “Effect of practice environment on nurse reported quality and patient safety: the mediation role of person-centeredness”.
Based on reviewers' suggestions in this section it has been pointed out that the sample was collected for convenience. Similarly, the information in the limitations section of the study has been expanded (Page 4, line 137).
In Data Analysis need to write the Abbreviations in full first time presented such as “QCA” or “fsQCA”.
According to reviewers' suggestions the full name has been included (Page 4, line 151).
Results:
Remove the explanations of Methods from results section. Move these to methodology.
As requested by the reviewers, the explanation of the steps performed in the regressions has been moved to the section of the method (Page 4, line 150-154).
Discussions:
Line 253: "it would be 251 interesting to explore the influence of other organizational variables such as type of con-252 tract, job satisfaction or school and work environment on the attitude towards communi-253 cation in nursing [42]"
Other factors also could knowledge self-efficacy, see this article :
- Shehab S, Al-Bsheish M, Meri A, Dauwed M, Aldhmadi BK, Kareem HM, Alsyouf A, Al-Mugheed K, Jarrar MT. Knowledge sharing behaviour among head nurses in online health communities: The moderating role of knowledge self-efficacy. PloS one. 2023 Jan 19;18(1):e0278721.
Following the reviewers' recommendations, the information on the suggested article has been added (Page 9, line 309-310).
.

Reviewer 2 Report
Dear Authors,
Thanks for the opportunity to review this interesting article. This article discusses the attitudes towards communication in nurses and nursing students. The authors make some useful points and could deliver contribution to the literature. I propose some revisions:
- In abstract section, last sentence, I propose you replace the period for a coma: “Our results suggest the importance of emotional intelligence, especially emotional repair and empathy, and lead us to consider…
- In Introduction section, last paragraph I suggest you change the sentence to: “Based on previous literature (H1) is expected that….Similarly (H2) we propose that lack of emotional…”
- In Materials and Methods Section, subsection Participants, you presented some characteristics of the participants. Do you asked for other characteristics as job conditions, professional experience, educational level?
- In subsection Procedure, you present the procedure for sample 1- professionals and sample 2 students, however in sample 1 you write that “The sample of nursing students was collected from five universities… Does the sample 1 was composed only for nurses or also for nursing students? Also , how you explain that in the sample 1 the time spent to answer the questionnaire was 35 min and in the sample 2 they completed the questionnaire in 10 min. Both samples haven’t answer to the same questionnaires? Why you collected the data in sample 2 two years after sample 1?
Could you please clarify this points?
- In subsection 3.3 line 188, I suppose you forgot some information, please correct this sentence.
- In discussion section, line 244 and 247 you write however. Could you please rewrite one of the sentences changing the word “However”.
Author Response
Dear editor and reviewers,
Thanking you in advance for your work, time and effort, and responding to your request and that of the reviewers, we are sending you the manuscript again. In it you can see in blue everything we have modified.
In summary, we have included more theoretical information in the introduction section and modified general aspects of the language. We have also provided more information in the method and discussion section and future lines of research.
Reviewer 2.
Dear Authors,
Thanks for the opportunity to review this interesting article. This article discusses the attitudes towards communication in nurses and nursing students. The authors make some useful points and could deliver contribution to the literature. I propose some revisions:
- In abstract section, last sentence, I propose you replace the period for a coma: “Our results suggest the importance of emotional intelligence, especially emotional repair and empathy, and lead us to consider…
Following the reviewers' recommendations, the period has been replaced by a comma (Page 1, line 26).
- In Introduction section, last paragraph I suggest you change the sentence to: “Based on previous literature (H1) is expected that….Similarly (H2) we propose that lack of emotional…”
As suggested by the reviewers the sentence has been modified (Page 2, line 89).
- In Materials and Methods Section, subsection Participants, you presented some characteristics of the participants. Do you asked for other characteristics as job conditions, professional experience, educational level?
- In subsection Procedure, you present the procedure for sample 1- professionals and sample 2 students, however in sample 1 you write that “The sample of nursing students was collected from five universities… Does the sample 1 was composed only for nurses or also for nursing students? Also, how you explain that in the sample 1 the time spent to answer the questionnaire was 35 min and in the sample 2 they completed the questionnaire in 10 min. Both samples haven’t answer to the same questionnaires? Why you collected the data in sample 2 two years after sample 1?
Could you please clarify this points?
As the reviewers pointed out, these aspects have been reviewed. Firstly, it has been clarified that the sample of professionals was only composed of professionals. Likewise, the information about the time taken by both samples to complete the questionnaires (35 minutes) has been modified (Page 4, line 137-138, 145).
Finally, it was included as a limitation of our study the fact that the sample was collected at two different points in time. This aspect was done this way for organizational reasons of the project and the institutions (Page 11, line 384-387).
- In subsection 3.3 line 188, I suppose you forgot some information, please correct this sentence.
As the reviewers highlighted it was a mistake, the missing information has been included (Page 7, line 239-240).
- In discussion section, line 244 and 247 you write however. Could you please rewrite one of the sentences changing the word “However”.
Following the suggestions of reviewers the word has been changed (Page 3, line 303).

Reviewer 3 Report
The subject of this manuscript is very relevant, both for the nursing profession and for the universities that train future nurses. As highlighted by the authors "The communication attitude (ACO) of nurses can significantly influence patient health outcomes".
The introduction very aptly situates the subject of the study.
Both the methodology and the results are clearly presented.
The discussion gives a very detailed account of the results and compares them with other similar research.
It is very opportune to highlight that modifiable variables such as social skills or emotional intelligence can influence attitudes towards communication, which can specifically focus training to improve ACO. It also stresses the need to establish training programmes for nurses in hospital centres.
I would only suggest to the authors to include a paragraph in which they elaborate a little more on how training to improve ACO would be included during the nursing degree.
Author Response
Dear editor and reviewers,
Thanking you in advance for your work, time and effort, and responding to your request and that of the reviewers, we are sending you the manuscript again. In it you can see in blue everything we have modified.
In summary, we have included more theoretical information in the introduction section and modified general aspects of the language. We have also provided more information in the method and discussion section and future lines of research.
Reviewer 3.
The subject of this manuscript is very relevant, both for the nursing profession and for the universities that train future nurses. As highlighted by the authors "The communication attitude (ACO) of nurses can significantly influence patient health outcomes".
The introduction very aptly situates the subject of the study.
Both the methodology and the results are clearly presented.
The discussion gives a very detailed account of the results and compares them with other similar research.
It is very opportune to highlight that modifiable variables such as social skills or emotional intelligence can influence attitudes towards communication, which can specifically focus training to improve ACO. It also stresses the need to establish training programmes for nurses in hospital centres.
I would only suggest to the authors to include a paragraph in which they elaborate a little more on how training to improve ACO would be included during the nursing degree.
According to reviewers' suggestions of this information has been included (Page 10, lines 366-373).

Round 2
Reviewer 2 Report
Dear authors,
Thanks for the opportunity to review this interesting article. I think you made the suggested changes, although i propose some minor changes,
In page 3, line 133 you presented the subtitle 2.3 twice, associated with procedure and data anlysis, please change this. In subtitle Procedure I propose you write the sentence"both professionals..." and not in professionals sample subsection.
In table 4, line 246 and line 239 please correct the word "necesary".
Author Response
Dear reviewer, thank you for your time and effort dedicated to this manuscript.
Your suggested changes have been incorporated and the manuscript has been fully revised.
Thank you again